# Linear Recurrent Networks Approximate Optimal Filtering in Hidden Markov Models

**Andrew Mah**
Flatiron Institute, Center for Computational Neuroscience
New York City, NY 10010
amah@flatiorninsitute.org

**Alex Williams**
Flatiron Institute, Center for Computational Neuroscience
New York University, Center for Neural Science
New York City, NY 10010
alex.h.williams@nyu.edu

## Abstract

Understanding the computational mechanisms by which neural networks perform probabilistic inference is a central problem in mechanistic interpretability and sequence modeling. Here, we use token sequences generated from Hidden Markov Models (HMMs) with known latent dynamics and analytically specified Bayes-optimal forward filters to provide a mechanistic understanding of probabilistic inference in recurrent neural networks. HMMs provide both a ground-truth generative process and an exact optimal solution, enabling precise comparison between model representations and optimal latent state inference. Surprisingly, purely linear recurrent networks with a softmax readout layer, either hand-engineered to approximate the optimal Bayesian filter or trained from data, achieve near-optimal prediction performance. Moreover, trained linear RNNs recover low-dimensional representations of latent state probabilities despite never being given direct access to those states. These findings suggest that linear recurrent architectures can serve as both effective and interpretable models for structured probabilistic sequence prediction.

## 1 Introduction

Understanding not only *what* types of problems neural networks are capable of solving, but also *how* networks are solving these problems, is a fundamental problem in computational neuroscience and mechanistic AI research [2, 3, 12]. Here, we focus on understanding sequential prediction tasks, which are fundamental to state-of-the-art generative language models. While modern architectures such as transformer-based models [1, 6, 14] and, more recently, recurrent state-space models [5, 8, 11] have achieved remarkable success across a variety of natural language tasks, interpreting these models is still incredibly difficult. Moreover, standard language tasks, such as next-word prediction, are themselves inherently difficult to interpret. There is rarely a single "optimal" solution and no universally accepted baseline for comparison [13].

In this work, we use Hidden-Markov models (HMMs) to define a ground-truth generative process for next-token prediction tasks. While simple, HMMs can, in principle, approximate any stationary process to high precision [4, 7, 9]. Furthermore, HMMs come equipped with a computationally efficient Bayes-optimal inference procedure (the forward filter [10]), enabling direct evaluation of a model's learned representations and their correspondence to optimal probabilistic inference.

Preprint.

Using sequences generated from HMMs, we reveal the surprising expressive capacity of purely linear recurrent networks with a softmax readout layer. We show that both hand-engineered linear networks designed to approximate optimal Bayesian filters and trained linear recurrent neural networks (RNNs) can achieve near-optimal performance on nontrivial sequence prediction tasks. Furthermore, we provide evidence that trained linear RNNs learn low-dimensional internal representations of the probabilistic latent states, even without directly observing these states during training. Our results highlight both the interpretability and effectiveness of linear RNNs in structured sequence prediction.

## 2 Preliminaries

### 2.1 Hidden-Markov Models

A discrete Hidden-Markov model (HMM) is a generative model in which observable emissions, $y_t \in \{1, ..., M\}$, depend on unobserved latent states, $x_t \in \{1, ..., N\}$. The distribution over emissions is determined by an emission matrix, $\boldsymbol{E} \in \mathbb{R}^{N \times M}$, where $E_{i,j} = p(y_t = j \mid x_t = i)$. Latent states evolve as a first-order Markov chain specified by a transition matrix, $\boldsymbol{T} \in \mathbb{R}^{N \times N}$, where $T_{i,j} = p(x_t = j \mid x_{t-1} = i)$. With sufficiently many states and emissions, HMMs can approximate any stationary stochastic process [4, 7, 9].

Given a sequence observations, $y_{1:t}$, the Bayes-optimal *forward filter* iteratively updates a posterior distribution over latent states, $\boldsymbol{\pi}_t$ by:

$$\pi_{t,i} = p(x_t = i \mid y_{1:t}) \propto p(y_t \mid x_t = i) \sum_j p(x_t = i \mid x_{t-1} = j) p(x_{t-1} = j \mid y_{1:t-1}). \quad (1)$$

Oftentimes, it is convenient to work with the latent state log-odds which are unbounded and do not require normalization at each step. For a fixed reference state, $N$, define the log-odds vector, $\boldsymbol{\alpha}_t \in \mathbb{R}^{N-1}$ by:

$$\alpha_{t,i} := \log \frac{\pi_{t,i}}{\pi_{t,N}}, \quad i = 1, ..., N - 1.$$

yielding a minimal representation, $\boldsymbol{\alpha}_t$, from which the full posterior can be recovered via a softmax operation. $\boldsymbol{\alpha}_t$ evolves according to a recursive update that separates the influence of observation likelihoods and dynamics induced by latent-state transitions.

**Lemma 2.1** (Recursive Update for Log-Odds). *The log-odds ratio satisfies*

$$\boldsymbol{\alpha}_t = \boldsymbol{\ell}_t + \boldsymbol{f}(\boldsymbol{\alpha}_{t-1})$$

*where:*

1. $\ell_{t,i} := \log \frac{p(y_t \mid x_t = i)}{p(y_t \mid x_t = N)}$ *is the observation log-likelihood ratio, and*

2. $f_i(\boldsymbol{\alpha}_{t-1}) : \mathbb{R}^{N-1} \to \mathbb{R}$, *is the log-ratio prior update function for state* $i$:

$$f_i(\boldsymbol{\alpha}_{t-1}) = \log \frac{T_{N,i} + \sum_{j=1}^{N-1} T_{j,i} \exp \alpha_{t-1,j}}{T_{N,N} + \sum_{j=1}^{N-1} T_{j,N} \exp \alpha_{t-1,j}}.$$

*Proof.* Using the recursive filter defined in Eq. 1, writing $\pi_{t,i} = \pi_{t,N} \exp(\alpha_{t,i})$, and simplifying the ratio $\alpha_{t,i} = \log(\pi_t(i)/\pi_t(N))$ yields the desired result. $\square$

This posterior directly yields the predictive distribution for the next observation by:

$$p(y_{t+1} \mid y_{1:t}) = \boldsymbol{E}^T \boldsymbol{T}^T \boldsymbol{\pi}_t. \quad (2)$$

### 2.2 Methods and Training Framework

To study the expressive capacity of linear RNNs, we hand-crafted HMM "teacher models" to generate long token sequences, $\boldsymbol{y}_{1:T}$, and trained linear RNNs to minimize the categorical cross-entropy between these tokens and RNN predictions, denoted $\hat{\boldsymbol{y}}_{1:T}$. Concretely, for each timestep $t$, let $\boldsymbol{h}_t$

denote the hidden-state activation of a linear RNN and $\boldsymbol{u}_t$ its inputs. The hidden-states are updated according to: $\boldsymbol{h}_t = \boldsymbol{W}_{\text{in}}\boldsymbol{u}_t + \boldsymbol{W}_{\text{rec}}\boldsymbol{h}_{t-1}$. Output logits for the next observation, $\hat{\boldsymbol{y}}_t$, are computed by a linear readout with softmax-activation: $\hat{\boldsymbol{y}}_t = \sigma(\boldsymbol{W}_{\text{out}}\boldsymbol{h}_t + \boldsymbol{b})$. We provide full experimental details in Appendix A.

## 3 Results

### 3.1 Linear RNNs perform near-optimally

We begin with a case study based on the classic Occasionally Dishonest Casino HMM (Fig. 1A, B). Results on additional HMM variants are summarized in Appendix B, and a theoretical discussion is provided in Section 3.4.

For this task, an agent observes the outcomes of die rolls in a casino that secretly switches between a fair die ($x_t = 1$) and a loaded die ($x_t = 2$) and must predict the next outcome. The HMM is defined by the following emission matrix, $E$, and transition matrix, $T$:

$$E = \begin{bmatrix} 1/6 & 1/6 & 1/6 & 1/6 & 1/6 & 1/6 \\ 1/10 & 1/10 & 1/10 & 1/10 & 1/10 & 1/2 \end{bmatrix}, \quad T = \begin{bmatrix} 0.95 & 0.05 \\ 0.1 & 0.9 \end{bmatrix}.$$

We trained a linear RNN with softmax output to predict the next-token distribution of the Occasionally Dishonest Casino. The model's outputs closely matched the Bayes-optimal next-token distribution on held-out test data (Fig. 1C). Unless otherwise noted, reported KL divergences are computed as $D_{KL}(p_{\text{Bayes}} \,\|\, p_{\text{model}})$, averaged over time steps and sequences. Model performance, measured by per-sequence perplexity (lower values indicate better performance), was nearly identical across models: the optimal filter achieved a score of $5.699 \pm 0.133$, the linear RNN $5.701 \pm 0.132$, and an LSTM baseline $5.699 \pm 0.134$.

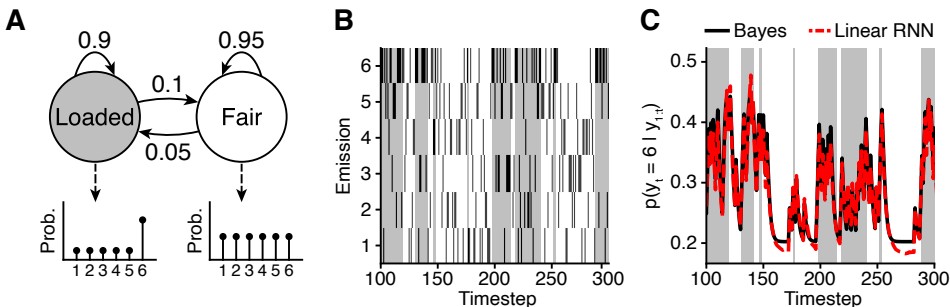

Figure 1: **Linear RNNs perform near-optimally on the Occasionally Dishonest Casino. A**. Schematic of the Occasionally Dishonest Casino. **B**. Example emissions sampled from A. **C**. Estimated probability of observing 6 of Bayesian forward filter (black) and linear RNN (red) on data from B.

### 3.2 Forward filter approximated by a linear dynamical system

We next examined how the linear RNN implements this computation. The exact recursive log-odds formulation from Lemma 2.1 includes a nonlinear prior update $\boldsymbol{f}$, which a linear network cannot directly represent. We demonstrate empirically and analytically (Section 3.4) that this function, however, can be well-approximated by a linear dynamical system. Expanding $\boldsymbol{f}$ around a reference point $\boldsymbol{\alpha}^*$ yields

$$\tilde{\boldsymbol{\alpha}}_t = \boldsymbol{\ell_t} + \boldsymbol{f}(\boldsymbol{\alpha}^*) + J_{\boldsymbol{f}}(\boldsymbol{\alpha}^*)(\tilde{\boldsymbol{\alpha}}_{t-1} - \boldsymbol{\alpha}^*), \tag{3}$$

where $\boldsymbol{f}$ and $\boldsymbol{\ell}_t$ are defined in Lemma 2.1 and $J_{\boldsymbol{f}}(\boldsymbol{\alpha}^*)$ is the Jacobian of $\boldsymbol{f}$ evaluated at $\boldsymbol{\alpha}^*$.

In the two-state Occasionally Dishonest Casino, this linearized filter reduces to a scalar recursion with a recurrent weight $f'(\alpha^*)$. We selected $\alpha^*$ to minimize the categorical cross-entropy on next-token predictions. The resulting hand-engineered linearized filter closely matches the exact forward filter (Fig 2A; MSE between $\alpha_t$ and $\tilde{\alpha}_t = 0.0901$), and produces similar next-token predictions (Fig 2B; mean KL divergence $= 3.823 \times 10^{-4}$, perplexity $= 5.701 \pm 0.129$). Specifically, we found that the linearized filter performs a leaky integration of evidence ($f'(\alpha^*) = 0.802 < 1$).

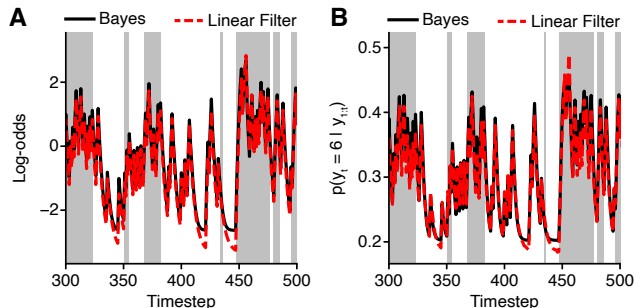

Figure 2: **Linearized filter well-approximates exact forward filter. A.** Log-odds from exact forward filter (black) and linearized forward filter (red). **B.** Estimated probability of observing 6 of Bayesian forward filter (black) and linearized filter (red).

### 3.3 Linearized forward filter explains RNN dynamics

Finally, we found that the hidden units of the linear RNN also implement a one-dimensional leaky-integrator, consistent with the linearized forward filter. The RNN's hidden-state activity was effectively one-dimensional, with the first principal component (PC) capturing over 98.8% of the variance (Fig. 3A). Surprisingly, the activity along the first PC was highly correlated with the log-odds from the linearized filter (Pearson $\rho = 0.996, p \ll 0.001$; Fig. 3B), despite never having access to the latent HMM states.

The recurrent weight matrix exhibited a single dominant real eigenvalue (0.768), indicating slow decay characteristic of leaky integration with a timescale of $\tau = -1/\log(0.768) \approx 3.9$ trials (Fig. 3C). This eigenvalue's corresponding eigenvector was nearly aligned with the first PC ($\theta = 0.0316$ radians). Input and output weights were similarly organized along this latent axis: the input vector for observing a 6 was roughly parallel to the first PC ($\theta = 0.0856$ rad), while others were anti-parallel (average $\theta = 2.90$ rad). Similarly, the readout vector for predicting a 6 was parallel ($\theta = 0.0917$ rad) while all other readout vectors were roughly anti-parallel (average $\theta = 2.85$ rad).

Together, these results show that despite only ever observing emissions from an HMM, the linear RNN learns a compact internal representation of the HMM's latent states, and can organize its input-output mappings around this representations to support an interpretable and near-optimal solution to the next-token prediction task.

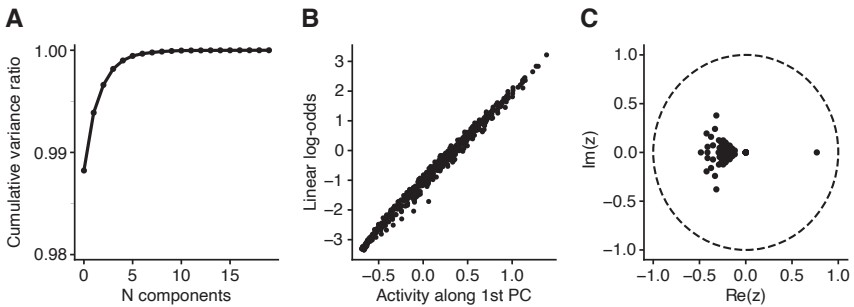

Figure 3: **Linear RNNs integrate evidence along low-dimensional subspaces. A**. PCA cumulative explained variance on hidden-unit activity **B.** Correlation between linear filter log-odds ratio and RNN activity along first PC. **C.** Eigenvalues of recurrent weight matrix.

### 3.4 Recursive Error Bound for Linearized Filters

Here, we derive mathematical results that apply to more general HMMs. Consider an HMM with $N$ latent states.

**Theorem 3.1** (Recursive Error Bound for Linearized Filters). *Let $\epsilon_t := \|\boldsymbol{\alpha}_t - \tilde{\boldsymbol{\alpha}}_t\|_2^2$ be the squared error between the Bayesian and linearized filter at time $t$. Then:*

$$\epsilon_t \leq 2 \left\| J_{\boldsymbol{F}}(\boldsymbol{\alpha}^*) \right\|_2^2 \epsilon_{t-1} + C \left\| \boldsymbol{\alpha}_{t-1} - \boldsymbol{\alpha}^* \right\|_2^4$$

*for some constant $C$ which depends on the second derivatives of $\boldsymbol{f}$.*

*Proof.* See Appendix C. □

This result provides conditions on the transition matrix, $\boldsymbol{T}$ to guarantee that the linear filter can closely approximate the true Bayesian log-odds. Specifically, if the latent-state Markov chain is irreducible and aperiodic (for example, if $T_{i,j} > 0$ for all $i, j$), then it admits a unique stationary distribution which corresponds to a fixed point in log-odds space. Choosing $\boldsymbol{\alpha}^*$ near this fixed point ensures that small deviations decay over time and the error recursion remains stable. Take together, these results provide a theoretical perspective that explains both how linear filters can be hand-engineered to approximate Bayesian filters and why linear RNNs are able to attain near-optimal filtering dynamics in practice.

## 4    Discussion

We have demonstrated that linear recurrent neural networks with a linear + softmax readout are capable of learning and representing nontrivial latent generative structures directly from sequential data.

### 4.1    Limitations

Importantly, our theoretical analysis (Theorem 3.1) shows that our linear approximation can, in principle, approximate Bayesian filters across a general classes of HMMs. However, our empirical validation is currently limited to small-scale, synthetic HMMs. Further work is required to demonstrate performance learning from higher-dimensional HMMs.

### 4.2    Broader Impact

This work introduces a principled framework for analyzing the probabilistic representations and inductive biases of trained neural networks. Future work extending the framework to empirical HMMs or to modern architectures such as state-space models and transformers could provide insight into the representational structure of state-of-the-art sequence models. Such advances can contribute to research in AI interpretability and safety alignment, with potential long-term benefits for developing more transparent and trustworthy machine learning systems.

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
