# A  Experimental Details

## A.1  Models

### A.1.1  Hand-engineered linear models

To generate hand-engineered linear models, for each HMM, we used the ground-truth emission matrix, $E$, and transition matrix, $T$. These matrices determine the observation log-likelihood ratio, $\ell_t$, and the log-ratio prior update function, $f$ as defined in Lemma 2.1. The only free parameter in this construction is the linearization point, $\alpha^*$, which we optimized to minimize the categorical cross-entropy between the model's predicted and true next-token distributions.

### A.1.2  Recurrent Neural Networks

All RNN models were implemented using the Equinox library [2].

Linear RNNs consisted of 64 hidden units and three components:

1. Input embedding: $W_{\text{in}} \in \mathbb{R}^{6\times 64}$, initialized row-wise from a standard Gaussian distribution and normalized to unit length.

2. Recurrent layer: $W_{\text{rec}} \in \mathbb{R}^{64\times 64}$, initialized as an orthogonal matrix scaled by 0.1.

3. Linear readout: $W_{\text{out}} \in \mathbb{R}^{64\times 6}$, with each row sampled from a uniform distribution over $\left[ -1/\sqrt{64}, 1/\sqrt{64} \right]$

LSTM models also used 64 hidden units and three components:

1. Input embedding: $W_{\text{in}} \in \mathbb{R}^{6\times 64}$, initialized row-wise from a standard Gaussian distribution and normalized to unit length.

2. Recurrent layer: a standard LSTM module with 64 cells using Equinox's default initialization.

3. Linear readout: $W_{\text{out}} \in \mathbb{R}^{64\times 6}$, with each row sampled from a uniform distribution over $\left[ -1/\sqrt{64}, 1/\sqrt{64} \right]$

## A.2  Training Procedures

Training was performed using the Adam optimizer from the Optax library [1]. For each HMM, we generated separate datasets for training (400,000 trials), validation (50,000 trials), and testing (50,000 trials). Each trial from the Occasionally Dishonest Casino datasets consisted of 512 observations, while trials from the Random HMM datasets contained 1,024 observations.

Training began with a learning rate of 0.001. The training data were divided into mini-batches of 64 trials, and the mean categorical cross-entropy loss with weight-decay (weights given in Table 1) was used to update model parameters. After each epoch, the validation loss was computed. If the validation loss failed to improve upon the previous minimum, the learning rate for the following epoch was reduced by half. Training was terminated once the validation loss failed to improve for three consecutive epochs.

Table 1: Regularization weight ($\log_{10}$ scale) for models across tasks for each model and task.

| Model | 1 dice | 2 dice | 3 dice | 4 dice | 5 dice | 6 dice | Random HMM |
|---|---|---|---|---|---|---|---|
| Linear | -0.889 | -1.556 | -1.815 | -1.938 | -7.000 | -7.543 | -3.000 |
| LSTM | -1.358 | -1.420 | -1.889 | -1.667 | -2.111 | -1.963 | -3.000 |

## A.3  Compute Resources

Models were fit on a NVIDIA RTX A6000 (48 GB VRAM, CUDA 12.8, driver 570.172.08). Each training run required between 0.5 and 1 hour on a single RTX A6000 GPU.

### A.4   Model performance metrics

#### A.4.1   Perplexity

For each model and test sequence, we computed the next-token perplexity as the exponential of the mean negative log-likelihood of the true observations under the model's predicted next-token distribution. Specifically, for a sequence of length T, perplexity, $P$ was given by

$$P = \exp\left(-\frac{1}{T}\sum_t p_{\text{model}}(x_t \mid x_{1:t-1})\right)$$

where $p_{\text{model}}(x_t \mid x_{1:t-1})$ denotes the model's next-token distibution evaluated on the following trial's observation. Reported values are averages over all test sequences.

#### A.4.2   KL-Divergence

For each model, we computed the mean Kullback-Leibler (KL) divergence between the Bayes-optimal next-token distribution $p_{\text{Bayes}}(x_{t+1} \mid x_{1:t})$ and the model's predicted distribution $p_{\text{model}}(x_{t+1} \mid x_{1:t})$ give by:

$$D_{KL}(p_{\text{Bayes}} \mid\mid p_{\text{model}}) = \frac{1}{T}\sum_t p_{\text{Bayes}} \log \frac{p_{\text{Bayes}}}{p_{\text{model}}}.$$

Reported values are averages over all test sequences.

### A.5   Code and Data Availability

The source code implementing all models, training, and evaluation will be released in a public repository upon publication of this paper. Training and testing datasets used in this project were generated from HMMs described above.

## B   Additional HMMs

To assess the generality of linear RNNs and linear filters on next-token prediction tasks, we trained models on two additional classes of HMMs.

### B.1   Occasionally Dishonest Casinos with Multiple Loaded Dice

We first considered a family of Occasionally Dishonest Casino HMMs with $N$ loaded dice, each favoring a different emission. For example, the variant with two loaded dice is defined by the following emission and transition matrices:

$$\boldsymbol{E} = \begin{bmatrix} 1/6 & 1/6 & 1/6 & 1/6 & 1/6 & 1/6 \\ 1/10 & 1/10 & 1/10 & 1/10 & 1/10 & 1/2 \\ 1/10 & 1/10 & 1/10 & 1/10 & 1/2 & 1/10 \end{bmatrix}, \quad \boldsymbol{T} = \begin{bmatrix} 0.95 & 0.025 & 0.025 \\ 0.1 & 0.9 & 0 \\ 0.1 & 0 & 0.9 \end{bmatrix}.$$

Each loaded die strongly biases one outcome while the fair die produces uniform emissions. As $N$ increases, the model must track a growing number of latent modes. Performance across these variants is summarized in Tables 2 and 3.

Table 2: Average Perplexity ($\pm$ standard deviation) on Dishonest Casino Variants with $N$ loaded dice

| $N$ | Bayes | Linear RNN | Linear Filter | LSTM |
|---|---|---|---|---|
| 1 | $5.699 \pm 0.133$ | $5.701 \pm 0.132$ | $5.701 \pm 0.129$ | $5.699 \pm 0.134$ |
| 2 | $5.763 \pm 0.119$ | $5.772 \pm 0.116$ | $5.767 \pm 0.114$ | $5.763 \pm 0.118$ |
| 3 | $5.795 \pm 0.112$ | $5.812 \pm 0.107$ | $5.800 \pm 0.106$ | $5.795 \pm 0.111$ |
| 4 | $5.816 \pm 0.107$ | $5.841 \pm 0.099$ | $5.821 \pm 0.101$ | $5.816 \pm 0.107$ |
| 5 | $5.831 \pm 0.103$ | $5.862 \pm 0.094$ | $5.836 \pm 0.098$ | $5.831 \pm 0.102$ |
| 6 | $5.842 \pm 0.101$ | $5.878 \pm 0.087$ | $5.847 \pm 0.096$ | $5.842 \pm 0.100$ |

Table 3: Average $D_{KL}(p_{\text{Bayes}} \parallel p_{\text{model}})$ on Dishonest Casino Variants with $N$ loaded dice

| N | Linear RNN | Linear Filter | LSTM |
|---|---|---|---|
| 1 | $(3.661 \pm 9.007) \times 10^{-4}$ | $(3.823 \pm 8.291) \times 10^{-4}$ | $(3.269 \pm 2.425) \times 10^{-5}$ |
| 2 | $(1.592 \pm 1.802) \times 10^{-3}$ | $(7.052 \pm 15.40) \times 10^{-4}$ | $(3.379 \pm 4.736) \times 10^{-5}$ |
| 3 | $(2.999 \pm 2.922) \times 10^{-3}$ | $(8.481 \pm 18.34) \times 10^{-4}$ | $(5.048 \pm 7.082) \times 10^{-5}$ |
| 4 | $(4.287 \pm 4.161) \times 10^{-3}$ | $(9.176 \pm 19.75) \times 10^{-4}$ | $(6.192 \pm 7.944) \times 10^{-5}$ |
| 5 | $(5.337 \pm 5.209) \times 10^{-3}$ | $(9.532 \pm 20.56) \times 10^{-4}$ | $(5.645 \pm 9.161) \times 10^{-5}$ |
| 6 | $(6.267 \pm 6.787) \times 10^{-3}$ | $(9.715 \pm 21.10) \times 10^{-4}$ | $(8.247 \pm 14.14) \times 10^{-5}$ |

Table 4: Model performance on 5 randomly sampled HMMs

| | Perplexity | | | $D_{KL}(p_{\text{bayes}} \parallel p_{\text{model}})$ | |
|---|---|---|---|---|---|
| | Bayes | Linear RNN | Linear Filter | Linear RNN | Linear Filter |
| HMM 1 | 9.759 | 9.759 | 9.760 | $3.289 \times 10^{-5}$ | $1.636 \times 10^{-4}$ |
| HMM 2 | 9.828 | 9.829 | 9.831 | $3.950 \times 10^{-5}$ | $2.251 \times 10^{-4}$ |
| HMM 3 | 2.487 | 9.515 | 9.516 | $2.487 \times 10^{-5}$ | $1.118 \times 10^{-4}$ |
| HMM 4 | 9.710 | 9.711 | 9.711 | $2.633 \times 10^{-5}$ | $1.137 \times 10^{-4}$ |
| HMM 5 | 9.697 | 9.697 | 9.698 | $2.847 \times 10^{-5}$ | $8.588 \times 10^{-5}$ |

## B.2 Randomly Generated HMMs

To further test generalization, we trained on a collection of randomly generated HMMs with 10 latent states and 15 emission types. For each model, the rows of $T$ and $E$ were independently sampled from Dirichlet distributions with uniform concentration parameters:

$$\boldsymbol{\alpha}_T = \mathbf{1} \in \mathbb{R}^{10}, \quad \boldsymbol{\alpha}_E = \mathbf{1} \in \mathbb{R}^{15}$$

Performance on these random HMMs is summarized in Table 4.

## C  Proofs

### C.1  Proof of Theorem 2.2

**Theorem C.1** (**Recursive Error Bound for Linearized Filters**). *Let $\epsilon_t := \|\boldsymbol{\alpha}_t - \tilde{\boldsymbol{\alpha}}_t\|_2^2$ be the squared error between the Bayesian and linearized filter at time $t$. Then:*

$$\epsilon_t \le 2 \|J_{\boldsymbol{f}}(\boldsymbol{\alpha}^*)\|_2^2 \epsilon_{t-1} + C \|\boldsymbol{\alpha}_{t-1} - \boldsymbol{\alpha}^*\|_2^4$$

*for some constant $C$ which depends on the second derivatives of $\boldsymbol{f}$.*

*Proof.* For each $i \in \{1, ..., N-1\}$, Taylor's theorem gives:

$$f_i(\boldsymbol{\alpha}_{t-1}) = f_i(\boldsymbol{\alpha}^*) + \nabla^T f_i(\boldsymbol{\alpha}^*)(\boldsymbol{\alpha}_{t-1} - \boldsymbol{\alpha}^*) + R_{t,i}$$

where the remainder term is given by

$$R_{t,i} = \frac{1}{2}(\boldsymbol{\alpha}_{t-1} - \boldsymbol{\alpha}^*)^T H_{f_i}(\boldsymbol{\xi}_{t,i})(\boldsymbol{\alpha}_{t-1} - \boldsymbol{\alpha}^*)$$

for some $\boldsymbol{\xi}_{t,i}$ on the line segment between $\boldsymbol{\alpha}_{t-1}$ and $\boldsymbol{\alpha}^*$. Using this expansion, the error becomes

$$\epsilon_t = \sum_i (f_i(\boldsymbol{\alpha}_{t-1}) - f_i(\boldsymbol{\alpha}^*) - \nabla^T f_i(\boldsymbol{\alpha}^*)(\tilde{\boldsymbol{\alpha}}_{t-1} - \boldsymbol{\alpha}^*))^2$$

$$\le 2 \sum_i (\nabla^T f_i(\boldsymbol{\alpha}^*)(\boldsymbol{\alpha}_{t-1} - \tilde{\boldsymbol{\alpha}}_{t-1}))^2 + 2 \sum_i R_{t,i}^2$$

$$= 2 \|J_{\boldsymbol{f}}(\boldsymbol{\alpha}^*)\|_2^2 \epsilon_{t-1} + 2 \sum_i R_{t,i}^2$$

One can show $H_{f_i}(\boldsymbol{x})$ is bounded for all $i$, so let $H_{\max} := \max_i \sup_{\boldsymbol{x}} H_{f_i}(\boldsymbol{x})$, then $R_{t,i} \leq \frac{1}{2} H_{\max} \|\boldsymbol{\alpha}_{t-1} - \boldsymbol{\alpha}^*\|_2^2$ and thus

$$\sum_i R_{t,i}^2 \leq \frac{(N-1)^2}{4} H_{\max} \|\boldsymbol{\alpha}_{t-1} - \boldsymbol{\alpha}^*\|_2^4.$$

Finally,

$$\epsilon_t \leq 2 \|J_{\boldsymbol{f}}(\boldsymbol{\alpha}^*)\|_2^2 \, \epsilon_{t-1} + C \|\boldsymbol{\alpha}_{t-1} - \boldsymbol{\alpha}^*\|_2^4$$

as claimed. $\qquad\square$