# OpenReview forum: "Linear Recurrent Networks Approximate Optimal Filtering in Hidden Markov Models"
_NeurIPS.cc/2025/Workshop/UniReps — UniReps2025_

### Official Review · Reviewer_dS1m · 2025-09-05
**Linear Recurrent Networks Approximate Optimal Filtering in Hidden Markov Models**

**Confidence:** 5

**Review:**

The title is not properly arranged, I will prefer it to be "Approximating Optimal Filtering in Hidden Markov Models with Linear Recurrent Neural Architectures"

The abstract is overly descriptive and lacks a clear focus on the key findings and contributions of the study. 1-15

The introduction is focussed, but fails to clearly state the research gap the study aims to fill.17-35

The review of related works was left out for in-depth and critical analysis.

Result session should not come after introducing the topic. 38

The discussion section is underdeveloped. It does not effectively explained the outcome of the research and no comparison with the existing study since there was no related works.  118-119

The research presented is sound and well-executed. Provided that all suggested corrections are adequately addressed, the work meets the standards required for publication.

**Score:**

4

**Topic Fit:**

2

---

### Official Review · Reviewer_zM3N · 2025-09-15

**Confidence:** 3

**Review:**

**Summary**

The authors propose to use discrete HMMs as data-generating processes (DGP) that can serve as a benchmark with known ground-truth labels and latent states for sequential modeling tasks. The authors then present a case study with linear RNNs trained to mimic the ground-truth DGP.

**Strengths**

- Benchmarking both the predictive and representational fidelity of sequence models is an important task and using discrete HMMs to generate such benchmarks is a potentially interesting direction.
- The paper is generally well-written and easy to follow. The theoretical results can be used to bound prediction error in linear dynamic models.

**Weaknesses**

- It would be important to narrow down the claims in the paper, especially since a single case study was preformed. Clearly, linear RNNs will not perform well on more complex tasks. I don't think that linear RNNs are ever used in practice for predictive tasks, but maybe they are useful as computational models of cognition (the authors should elaborate). Maybe it's just me, but to me the empirical results are hardly surprising: a linear RNN was able to approximate a linear problem.
- It would have been interesting to compare the RNN results to those obtained with a similarly simple transformer architecture (e.g., linear attention) and relate to the theoretical results.

**Minor**
There are some typos (e.g., L113) and the extended abstract can profit from some proofreading.

**Score:**

2

**Topic Fit:**

2

---

### Official Review · Reviewer_uJLE · 2025-09-16
**Clear claim but limited and narrow comparison**

**Confidence:** 5

**Review:**

The paper studies next-token prediction on sequences generated by discrete HMMs and claims that purely linear RNNs with a softmax readout can achieve near–Bayes-optimal performance and learn low-dimensional internal representations of latent posteriors. It presents (i) a case study on the “Occasionally Dishonest Casino” HMM, (ii) PCA analyses of hidden states, and (iii) a linearization of the HMM forward filter with an error recursion bound (Theorem 2.2)

Strengths

The paper is clear written and self-contained using HMMs where an optimal inference procedure exists, making comparisons well-posed. The author uses a linear RNN and softmax that tracks Bayes predictions closely on the 2-state casino example, PCA shows a 1-D manifold aligned with log-odds. The reproducibility details are provided (optimizer, sequence lengths, batch sizes, init, hardware).

Weaknesses

The scope of experiments is extremely narrow. Results are effectively a single case study on the 2-state casino HMM. There is no systematic sweep over states N, emissions M, mixing regimes, near-degenerate emissions, or transition structures. Claims of “near-optimal” behavior are not stress-tested across families of HMMs.

The paper misses comparisons to classic linear/probabilistic baselines and modern sequence models. The work does not compare against: (i) direct implementations of the optimal forward filter as a learned linear state-space model; (ii) predictive-state representations / spectral learning for HMMs; (iii) linear dynamical models where emissions are categorical; (iv) simple nonlinear RNNs/GRUs; (v) SSMs (even a tiny Mamba-like toy) trained on the same sequences. Without these, it’s impossible to judge whether the linear RNN is uniquely effective/interpretable versus other almost-linear formalisms.
The paper states near-optimal performance but reports only average test losses with mean±sd on a single HMM and relatively small test sets; no calibration, KL to the Bayes posterior, or regret curves versus horizon.

Summary

The paper is too narrow empirically, theoretical contributions are light, and comparisons are insufficient, so I suggest to reject.

**Score:**

2

**Topic Fit:**

3